# miR-193b-3p Promotes Proliferation of Goat Skeletal Muscle Satellite Cells through Activating IGF2BP1

**DOI:** 10.3390/ijms232415760

**Published:** 2022-12-12

**Authors:** Li Li, Xiao Zhang, Hailong Yang, Xiaoli Xu, Yuan Chen, Dinghui Dai, Siyuan Zhan, Jiazhong Guo, Tao Zhong, Linjie Wang, Jiaxue Cao, Hongping Zhang

**Affiliations:** Farm Animal Genetic Resources Exploration and Innovation Key Laboratory of Sichuan Province, College of Animal Science and Technology, Sichuan Agricultural University, Chengdu 611130, China

**Keywords:** miR-193b-3p, insulin-like growth factor-2 mRNA-binding protein 1 (IGF2BP1), transcriptional activation, myogenic differentiation

## Abstract

As a well-known cancer-related miRNA, miR-193b-3p is enriched in skeletal muscle and dysregulated in muscle disease. However, the mechanism underpinning this has not been addressed so far. Here, we probed the impact of miR-193b-3p on myogenesis by mainly using goat tissues and skeletal muscle satellite cells (MuSCs), compared with mouse C2C12 myoblasts. miR-193b-3p is highly expressed in goat skeletal muscles, and ectopic miR-193b-3p promotes MuSCs proliferation and differentiation. Moreover, insulin-like growth factor-2 mRNA-binding protein 1 (IGF2BP1) is the most activated insulin signaling gene when there is overexpression of miR-193b-3p; the miRNA recognition element (MRE) within the IGF1BP1 3′ untranslated region (UTR) is indispensable for its activation. Consistently, expression patterns and functions of IGF2BP1 were similar to those of miR-193b-3p in tissues and MuSCs. In comparison, ectopic miR-193b-3p failed to induce PAX7 expression and myoblast proliferation when there was IGF2BP1 knockdown. Furthermore, miR-193b-3p destabilized IGF2BP1 mRNA, but unexpectedly promoted levels of IGF2BP1 heteronuclear RNA (hnRNA), dramatically. Moreover, miR-193b-3p could induce its neighboring genes. However, miR-193b-3p inversely regulated IGF2BP1 and myoblast proliferation in the mouse C2C12 myoblast. These data unveil that goat miR-193b-3p promotes myoblast proliferation via activating IGF2BP1 by binding to its 3′ UTR. Our novel findings highlight the positive regulation between miRNA and its target genes in muscle development, which further extends the repertoire of miRNA functions.

## 1. Introduction

Formation of skeletal muscle (myogenesis) is a complex program covering the embryonic stage driven by myogenic progenitor cells [1], and the postnatal phase mainly contributed to by skeletal muscle satellite cells (MuSCs). MuSCs function through sequential progress, including proliferation, differentiation, and fusion of myoblasts [2], in which several protein-coding genes, such as paired box 7 (PAX7), myogenic differentiation protein 1 (MyoD), myogenin (MyoG) [3], and insulin-like growth factor family [4], play critical roles in myogenesis. Additionally, several muscle-specific microRNAs (miRNAs), including miR-1/miR-206 and miR-133 families [5,6], as well as a few ubiquitously expressed miRNAs such as miR-125b [7] and miR-223 [8], play essential roles in myogenesis.

Typically, miRNAs negatively regulate gene expression through miRNA-mRNA interaction [9], e.g., miRNAs bind complementary target RNAs and block translation (11–16%) t [10] or trigger the destruction of targeted mRNA (a majority, >84%) [11]. Nevertheless, accumulating evidence support that cytoplasmic miRNAs can also activate gene expression unconventionally through enhancing the translation by seed-matching binding on the 5′ [12] or 3′ untranslated region (UTR) [13], or even non-seed-matching sites in the 5′ UTR of mRNAs [14]. Meanwhile, a few studies suggest that nucleus-located miRNAs mediate gene activation epigenetically via targeting enhancers [15] or targeting the 5′ -end of a gene [16]. Undoubtedly, the functional significance of the activating miRNAs remains largely undetermined.

As a well-accepted anti-tumor oncogene, miR-193b generally targets and negatively regulates genes involving the cell cycle [17,18], whether miR-193b is markedly downregulated [19] or dramatically elevated in some cancers [20,21]. Outstandingly, miR-193b is usually dysregulated in muscle diseases such as Duchenne muscular dystrophy (DMD) [22,23] and governs the transition between myoblasts and brown adipocytes [24,25]. These suggest that the function of miR-193b-3p in myogenesis remains to be systematically illustrated.

Here, mainly using tissues and MuSCs from goats, we found out that miR-193b-3p was enriched in skeletal muscle. Furthermore, the insulin-like growth factor-2 mRNA-binding protein 1 (IGF2BP1), an anti-apoptotic gene that certainly promotes cell proliferation [26,27], was targeted and activated by miR-193b-3p transcriptionally via seed-matching sites in the 3′ UTR. In addition, the miR-193b-3p /IGF2BP1 axis mainly promoted myoblast proliferation. Our research provides a novel mechanism of miRNA activating its target gene transcriptionally.

## 2. Results

### 2.1. miR-193b-3p Promotes Myogenesis of Goat MuSC

According to the human genome (GRCh38/hg38, UCSC Genome Browser), miR193BHG (MIR193B host gene) is located on human Chr16 and exhibits high sequence conservation among species. Notably, the 22-nt nucleotide sequence of miR-193b-3p was identical among goat, mouse, rat, bovine, and even chicken genomes, whereas it differed in humans (the 10th A/T) (Figure 1A). Although several studies unveil the adipogenesis-inducing function of miR-193b signaling [28,29] and anti-tumor regulation of miR-193b in many cancers [19,30,31], levels of MIR193BHG in human skeletal muscle ranked second among 53 tissues (Figure 1B, upper panel; Appendix A). In particular, hsa-miR-193b-3p is more dominant than hsa-miR-193b-5p in muscles (Figure 1B, lower panel).

Using RNAs extracted from goat tissues and cultured MuSCs, we also found out that compared with other internal organs, extinguished miR-193b-3p levels were presented in goat muscles from 75 d embryos (E75) and newborn goats (Figure 1C). In addition, miR-193b-3p abundance continually increased in LD muscles with goat development (Figure 1C). Moreover, miR-193b-3p was dramatically elevated when shifting MuSC from GM to DM (Figure 1C). These suggest that miR-193b-3p most likely plays a critical role in mammal skeletal muscle development.

To address the function of miR-193b-3p in muscle, we first transfected miR-193b-3p mimic (mi-193b-3p, 5 μg/mL) into goat MuSC cultured in vitro. We found out that compared with the control (mi-Ctrl), ectopic miR-193b-3p (~170 foldchange) greatly upregulated mRNA levels of proliferation and differentiation myogenic genes, including PAX7, PCNA, MyoG, MyHC, and myomaker (Figure 2A). Furthermore, protein levels of PAX7 and MyoG quantified via western blotting (WB) confirmed their elevation caused by miR-193b-3p (Figure 2B). Additionally, we employed CCK-8 and EdU assay to monitor the proliferation ability and unveiled that miR-193b-3p mimic transfected for 24 h to 72 h greatly upregulated OD value at 450 nm (Figure 2C), as well as the EdU^+^ cells at 48 h (Figure 2D,E). Moreover, the myotube formation was induced by miR-193b-3p mimic, as shown by the increased size/number of MyHC staining cells at d 7 of differentiation (Figure 2D, right panel). Contrarily, inhibiting miR-193b-3p (in-193b-3p, 100 nmol) downregulated myogenic gene expression by 30~50% (*p* > 0.05, Figure 2F,G), compared to the control (in-Ctrl). Consistently, myoblast proliferation and myotube formation were also retarded by deficiency of miR-193b-3p, as shown by OD values (Figure 2H), the fold change of EdU^+^ cells, and MyHC staining myotubes (Figure 2I,J).

Factors affecting cell viability, including autophagy and apoptosis, also play a critical role in cell proliferation and differentiation [32]. Therefore, we evaluated the mRNA level of the essential autophagy-related gene LC3B (Microtubule Associated Protein 1 Light Chain 3 Beta) [33] and BECN1 (Beclin 1), as well as apoptosis-related Caspase1 [34]. We found out that levels of BECN1 transcripts were elevated by ~1.5 fold when overexpressing miR-193b-3p (*p* < 0.05), while others were unaffected by perturbing miR-193b-3p (*p* > 0.05) (Appendix A).

These results imply that miR-193b-3p closely regulates the myogenesis of MuSC, which may be mainly attributed to the activation of proliferation ability.

### 2.2. miR-193b-3p Targets and Induces IGF2BP1 Expression

To systematically anchor the overall effect of miR-193b-3p on global gene expression in the myoblast, we performed transcriptome profiling in MuSCs treated with miR-193b-3p mimic and control (*n* = 4). Total RNA from mycoplasma-free cells was extracted, qualified (Appendix A), and then sequenced to analyze protein-coding. Over 8.29 Gb clean bases were identified for each sample, and the total mapped reads and unique mapped reads were as high as 97% and 94%, respectively (Appendix A). PCA analysis suggests that these samples were expectedly distinguished into two groups (Appendix A).

Among the detected 20,268 transcripts, 471 were differentially expressed (DE, Padj < 0.05, Appendix A). The functional analysis suggested that the upregulated DE genes (*n* = 264) were significantly enriched in cell proliferation and muscle development (Appendix A). Among these, insulin-like growth factor binding was noticeable (GO:0005520, Padj = 0.015, Appendix A), since insulin-like growth factors (IGFs) play critical roles in muscle hypertrophy and regeneration through interplaying with myogenic regulators such as MyoD and MyoG [35]. We analyzed all 17 insulin family genes detected and found out that IGF2BP1 was the most dramatically affected one out of 14 genes induced by miR-193b-3p (Appendix A).

Moreover, we validated the transcript levels using qPCR (Appendix A and Figure 3A). Compared with IGF2 and IGF1R, IGF2BP1 transcripts were remarkably upregulated when overexpressing miR-193b-3p, while decreased when inhibiting miR-193b-3p (Figure 3A). Consistently, IGF2BP1 protein levels were significantly elevated by miR-193b-3p mimic (Figure 3B). Additionally, similar to the miR-193b-3p expression profile, IGF2BP1 highly accumulated in goat skeletal muscles from E75 to newborn (Figure 3C) and increased when MuSC shifted to differentiation (Figure 3B). These suggest that miR-193b-3p potentially activates expression of IGF2BP1.

Generally, miRNAs regulate gene expression through miRNA–mRNA interaction through sequence complementarity [9]. To validate the association between miR-193b-3p and IGF2BP1, first of all, by using online software FINDTAR3 version 1.0 and BIBIServ2, we found out that IGF2BP1 was the most excellent gene targeted by miR-193b-3 with shallow free energy (Mfe = −32.50 kcal/mol) (Figure 3D). To confirm the importance of base pairing between miR-193b-3p and IGF2BP1, we constructed the wild-type vectors (GGCCGG, R-Luc-IGF2BP1-wt) and the mutated ones (GTAAAG, R-Luc-IGF2BP1-mut) and then transfected them into MuSC separately. As shown in Figure 3E, without miR-193b-3p, luciferase activity maintained similar among groups, including R-Luc-NC, R-Luc-IGF2BP1-wt, and R-Luc-IGF2BP1-mut. In contrast, the addition of miR-193b-3p mimics the powerfully activated luciferase activity of R-Luc-IGF2BP1-wt (*p* < 0.001), but causes an insignificant change in R-Luc-IGF2BP1-mut bearing seed-site mutations (Figure 3E). Furthermore, in HeLa and C2C12 cells, we confirmed that 193b-3p substantially activated RLuc-IGF2BP1-wt luciferase activity, compared to the RLuc-NC (Figure 3F). These results imply that IGF2BP1 is directly targeted and activated by miR-193b-3p.

### 2.3. IGF2BP1 Intermediates Function of MiR-193b-3p on Myoblast Proliferation

It was previously reported that IGF2BPs (also named IMPs) participate in the development of many tissues through cell proliferation, especially HMGA2–IGF2BP2 axis is critical in regulating the activation of satellite cells toward myogenesis [36]. To address the function of IGF2BP1 in myogenesis, we constructed overexpressing vectors using the CDS of IGF2BP1 (pIGF2BP1, 3 μg/mL) and designed small interfering RNAs against IGF2BP1 (siIGF2BP1, 100 nM), and transfected them into MuSC separately. As shown in Figure 4A, pIGF2BP1 dramatically promoted its mRNA (*p* < 0.001) and protein levels, while siIGF2BP1 resulted in ~20% downregulation of IGF2BP1 mRNA (*p* < 0.01) but no dominant protein density was altered, which likely resulted from the low endogenous IGF2BP1 in cells. Additionally, pIGF2BP1-treated cells contained higher transcripts of PAX7, MyoD, and MyoG. Conversely, deficiency of IGF2BP1 (siIGF2BP1) was accompanied by downregulation of all myogenic genes (*p* > 0.10) (Figure 4B). Correspondingly, the protein levels of PAX7 and MyoG elevated in pIGF2BP1-treated MuSCs coincided with their mRNA abundance (Figure 4C), and EdU^+^ cells were increased in pIGF2BP1 and decreased in siIGF2BP1-treated cells, compared to the control group (Figure 4D). Moreover, we simply discriminated sub-cycling cells using flow cytometric analysis. The pIGF2BP1 treatment upregulated S-phase cells but downregulated G1/G0 cells (Figure 4D). These results phenocopy overexpression or deficiency of the miR-193b phenotype, respectively.

Furthermore, to validate whether function of miR-193b-3p is mediated by IGF2BP1, we transfected miR-193b-3p mimic (mi-193b-3p) and simultaneously interfering IGF2BP1 (siIGF2BP1) in MuSC. As shown in Figure 4E, IGF2BP1 deficiency neutralized the abundance of miR-193b-3p-elevated proliferation genes, including PAX7 and PCNA (*p* > 0.10), but not differentiation genes including MyoD, MyoG, and Myomaker (*p* < 0.05). Furthermore, comparing with the control, the fold change of EdU^+^ cells at 24 h, 48 h, and 72 h were insignificantly altered in cotransfected cells (Figure 4E), suggesting that instead of myogenic differentiation, the IGF2BP1 is more likely mediating function of miR-193b-3p in myoblast proliferation.

### 2.4. Goat miR-193b-3p Activates IGF2BP1 Transcription

To address the spatial localization of miR-193b-3p and IGF2BP1 mRNA in cells, we designed fluorescence-labeled oligonucleotide probes and performed RNA-RNA dual-labeled fluorescence in situ hybridization (FISH). Transcripts of miR-193b-3p and IGF2BP1 were dominantly overlapped in the cytoplasm of MuSC culturing, in growth (GM 48h) and differentiation media (DM 7d, Figure 5A). Typically, over 84% of miRNAs bind complementary target RNAs and trigger their destruction [11]. To monitor the effect of miR-193b-3p on turnover of endogenous IGF2BP1 mRNA, we treated culturing MuSCs with actinomycin D (ActD, 5 μg/mL) to compromise transcription effectively [37]. In addition, we then quantified the remaining levels of IGF2BP1 transcripts at three sequential time points. Expectedly, miR-193b-3p mimic dramatically degraded IGF2BP1 transcripts, in particular at 120 min (*p* < 0.01). In contrast, interfering miR-193b-3p exhibited no effect on the stability of IGF2BP1 mRNAs (Figure 5B). These suggest that miR-193b-3p destabilizing IGF2BP1 transcripts in the cytoplasm is similar to the typical function [38], and otherwise the mechanism implicates promoting IGF2BP1.

Currently, numerous mature miRNAs are identified in the nucleus [39], accompanied by several models unveiling transcriptional gene activation (TGA) closely related to the promoter and enhancer [15,39,40]. To explore whether miR-193b-3p functions in the nucleus, first of all, we quantified mature miR-193b-3p in RNA samples separately extracted from the cytoplasm and nucleus of GM and DM myoblasts. Comparing with U6 snRNA (mainly nucleus-located) and 18s rRNA (dominant in the cytoplasm), a small part of miR-193b-3p was located in the nucleus, especially in GM myoblasts (GM 7.56 ± 0.91% vs. DM 2.73 ± 0.54%, *p* < 0.01) (Figure 5C). Furthermore, we found out that ectopic miR-193b-3p (mi-193b-3p) efficiently unregulated IGF2BP1 hnRNA by ~4 fold as compared to its control (mi-Ctrl, *p* < 0.01), while inhibiting miR-193b-3p did not alter IGF2BP1 hnRNA significantly (Figure 5D). These imply that miR-193b-3p possibly stimulates IGF2BP1 transcription.

Furthermore, using data from UCSC, we found that the hsa-miR-193b region in human skeletal muscle myoblasts (HSMM) is enriched with the H3K4Me3 and H3K27Ac histone mark (Figure 5E upper panel, and Appendix A), a signal used for a promoter and active enhancer discovery, respectively [41]. Whereas mouse miR-193b barely overlaps with eRNA markers (Appendix A). On the other hand, both human and mouse IGF2BP1 3′ UTR region lack histone modification (Appendix A), suggesting that the activation of miR-193b-3p on IGF2BP1 is most likely attributed to the enhancer-related characteristics of miR-193b-3p.

A mature miRNA and its passenger strand present synergistic function and the same phenotype [42]. Therefore, to validate whether the goat miR-193b gene is similar to enhancer-related hsa-miR-193b-3p, we amplified ~60 bp 5p and 3p region, inserted them into a PGL3-Promoter vector separately (named pre-193b-5p and pre-193b-3p, Figure 5E lower panel), and transfected these vectors into MuSC and HeLa. As shown in Figure 5F, comparing to the basic (pBasic) and control (pCtrl), pre-193b-3p efficiently promoted F-Luc/R-Luc levels in GM and DM myoblasts, as well as in HeLa cells (*p* < 0.01 or 0.001). Meanwhile, pre-193b-5p enhanced F-Luc/R-Luc values significantly in DM cells (*p* < 0.001). These imply that the sequence of goat miR-193b could activate gene expression.

Another characteristic of activation-related miRNAs is their cis-activation on neighboring genes [15]. According to the goat genome, the adjacent genes for miR-193b include MKL2 (MKL/Myocardin Like 2), PARN (Poly(A)-Specific Ribonuclease), and BFAR (Bifunctional Apoptosis Regulator) (Figure 5G). We therefore quantified transcripts of these genes in goat MuSCs after perturbing miR-193b-3p, and found that in GM myoblasts, their expression was markedly waved along with the miR-193b-3p levels; while in DM cells, miR-193b-3p changed these three genes modestly (Figure 5H). Furthermore, we constructed vectors overexpressing the pri-miR-193b region (~400 bp) and unveiled that IGF2BP1 and PARN mRNA were significantly increased by ectopic pri-miR-193b in GM cells (*p* < 0.01, Figure 5I). Additionally, disturbing miR-193b-3p failed to alter miR-365-3p in goat MuSCs (Figure 5J, left panel), inconsistent with the results reported in mouse C2C12 [25], which may be ascribed to the differed location of miR-365 and miR-193b among species (Figure 5J, right panel, and Appendix A).

Although the mature sequence and the neighboring genes of miR-193b-3p are identical in the goat and mouse (Figure 1A and Figure 5G), it was previously reported that ectopic miR-193b blocks the shift of mouse C2C12 toward myogenesis [25]. To validate the function of miR-193b-3p in mice, we transfected the miR-193b-3p mimic (100 nmol) into C2C12 and found out that, opposite to that in goats, miR-193b-3p decreased IGF2BP1 mRNA levels (*p* < 0.05); on the contrary, an inhibitor of miR-193b-3p increased transcripts of IGF2BP1 (*p* < 0.10; Figure 6A). Meanwhile, levels of cell number-related genes, including PCNA, cyclin D1, and caspase1 (Figure 6A) and the neighboring genes, changed modestly by altering miR-193b-3p (Figure 6B). Furthermore, EdU^+^ cells were also correspondingly downregulated in miR-193b-3p mimic treated cells or upregulated when there was miR-193b-3p deficiency. Accordingly, ectopic miR-193b-3p slightly elevated G1/G0 cells and decreased S-phage cells (Figure 6C), suggesting that miR-193b-3p exhibits no activation of IGF2BP1 in mouse myogenesis.

These imply that goat miR-193b-3p is most likely similar to has-miR-193b, working as an enhancer-related miRNA that activates the target gene transcriptionally.

## 3. Discussion

Muscle formation is precisely mastered by a handful of hierarchical gene cascades covering protein-coding genes such as MRFs, as well as non-coding genes such as miRNAs. It is well-accepted that miRNAs play critical roles in muscle development through inversely fine-tune gene enrichment, canonically [5,6]. Nevertheless, a few miRNAs have gained much attention for newly appreciated activation roles indirectly targeting genes [14,15]. Here, we reported that the anti-tumor gene miR-193b-3p induces proliferation and differentiation of goat MuSCs, which phenocopies IGF2BP1 overexpression. Meanwhile, IGF2BP1 knockdown impairs myogenic proliferation, but not differentiation, of MuSCs induced by miR-193b-3p supplementation. Furthermore, miR-193b-3p located in the nucleus, blooms IGF2BP1 abundance transcriptionally and its neighboring genes. In addition, miR-193b-3p destructs IGF2BP1 mRNA in the cytoplasm, which is in line with the regulation of miR-193b-3p on IGF2BP1 transcripts in mice C2C12 myoblasts, reported previously [25]. These results suggest that miR-193b-3p/IGF2BP1 axis activates goat myogenesis, mainly attributed to its nucleus function.

### 3.1. Anti-Proliferation vs. Pro-Proliferation Function of miR-193b in Muscle

In cancer tissues, miR-193b-3p functions oppositely as an anti-tumor [19,43] or tumor-inducer [44] under differing disease conditions. As for the normal development, miR-193b signaling induces adipogenesis for adipose-derived stem cells [28,29] or is even shifted from C2C12 cells [25]. On the contrary, deficiency of miR-193b induces cells from mouse BAT to the muscle lineage with upregulated levels of muscle-specific genes, including Cmk, Myf5, Myf6, MyoD, and MyoG [25]. Although muscle diseases such as Duchenne muscular dystrophy 1 (DMD1) and DMD2 share similar symptoms, including gradually worsening muscle loss and weakness, miR-193b is accumulating in DMD1 patients [22] but is downregulated in DMD2 [23], suggesting their miR-193b-related intrinsic mechanism differs. Using C2C12 myoblasts, we validated the suppressing effect of miR-193b-3p on myogenic proliferation reported previously [25]. On the contrary, the gene expression, CCK-8, EdU, and cell cycle analysis implied that miR-193b-3p induced goat MuSCs proliferation toward the myotube. These suggest that the myogenic function of miR-193b-3p is closely associated with the microenvironment and species.

### 3.2. Activation of miR-193b on IGF2BP1 Transcription

For a long time, miRNAs have been tacitly assumed as cytoplasm located and post-transcriptional negative regulators of its target gene expression [10,11]. With high-throughput assessment, numerous mature miRNAs are currently identified in the nucleus [39]. Correspondingly, several models elaborating functions of nuclear miRNAs at transcriptional gene silencing (TGS) and transcriptional gene activation (TGA) have been constructed. MiRNA targets and binds nascent RNA transcripts, gene promoter, or enhancer regions to exert different effects via recruiting additional epigenetic and (or) transcriptional factors [39]. For example, miR-483-5p promotes IGF2 transcription by targeting the P3 5′ UTR and promoter, and enriching enhancer marks such as H3K4me3 and H3K27ac [16].

Given the close relationship between gene enhancer and promoter [45,46], we speculate that most likely enhancer-related miRNA-target pair, including enhancer-related miRNA or enhancer-related target, play a critical role in the activation of miRNAs on its target. Currently, the enhancer-related miRNAs are classified into the super-enhancers (SEs)-miRNAs (SEs neighbor cell-type-specific abundant miRNAs) that boost pri-miRNA processing [47] and the enhancer-overlapped-miRNAs (transcribed from enhancer region) that activate their neighboring gene in cis, as well as global genes in trans [15]. Here, we validated that goat miR-193b-3p induced IGF2BP1 through the complementary binding on its 3′ UTR. Furthermore, goat miR-193b-3p was partially nucleus-distributed and promoted IGF2BP1 hnRNA. In addition, both pre-miR193b-5p and 3p could activate gene expression and promote its neighboring genes’ expression, especially in GM MuSCs. Nevertheless, in mouse C2C12 myoblasts, miR-193b-3p negatively regulated IGF2BP1, which may be ascribed to the absence of H3K27ac and H3K4Me3 modification on the mouse miR-193b-3p gene locus. Additionally, the differed character between goat myoblasts and immortalized C2C12 could potentially affect their effect on miR-193b-3p function, similar to the inconsistent apoptotic responses between cell lines and primary culture cells [48].

These suggest that goat miR-193b-3p is most likely similar to hsa-193b-3p, which works as an enhancer-overlapped miRNA in the nucleus.

### 3.3. Other Factors Possibly Involved in miR-193b-3p/IGF2BP1 Axis

IGFs play critical roles in myogenesis because they can uniquely promote muscle cell proliferation (IGFs-SHC-RAS-MEK-ERK1/2) and differentiation (IGFs-IRS1-PI3K-AKT-Foxo3a), and induce MRFs mutually [35]. In goat MuSCs treated with miR-193b-3p mimic, IGF2BP1 was the most affected IGFs gene. IGF2BP1 promotes proliferation in mouse embryonic fibroblasts [49]. Similarly, we found out that IGF2BP1 could induce myogenic proliferation and differentiation of goat MuSCs, while results from cotransfection imply that miR-193b-3p/IGF2BP1 axis mainly affects proliferation. It is worth noting that IGF2BP1 stabilizes plentiful mRNA transcripts in an m^6^A-dependent manner [27]. Meanwhile, ectopic miR-193b-3p upregulated expression of METTL3 (*p* value = 0.017), an m(6)A methyltransferase critical for muscle growth [50], through m6A RNA methylation/YTHDF1/2 signaling axis [51] and interacting with MRFs [52,53]. These suggest that m^6^A might be involved in the miR-193b-3p/IGF2BP1 pathway in promoting myoblast proliferation. Moreover, given the many-to-many [54] and stage-specific [8] targeting manners between, as well as the RNA A-to-I editing modification of miR-193b-3p [55], future studies should also be conducted to explore the function of these factors in the miR-193b-3p/IGF2BP1 pathway.

## 4. Materials and Methods

### 4.1. Ethics Declarations

All experimental protocols in this study were strictly conducted following the Regulations for the Administration of Affairs Concerning Experimental Animals by China Ministry of Science and Technology and entirely approved by the Animal Care and Use Committee, Sichuan Agricultural University (No. DKY-20190035).

### 4.2. Animals and Samples Collection

A total of 12 goat fetuses and kids aged 45, 60, and 105 days of gestation and 3 days postnatal (*n* = 3 per stage; female) were randomly chosen and humanely sacrificed at the Jianzhou Big-Eared goats farm (Chengdu, China). Samples from the heart, liver, lung, spleen, kidney, and longissimus dorsi (LD) muscle were quickly collected and snap-frozen in liquid nitrogen for further study.

### 4.3. Isolating and Validating Goat MuSCs

We successfully isolated MuSCs from the LD muscles of neonatal goats [56]. Briefly, LD muscle blocks were washed with sterile phosphate-buffered saline (PBS, Hyclone, Logan, UT, USA), minced with ophthalmic scissors, digested in 0.2% Pronase at 37 °C for 1 h (Sigma-Aldrich), and centrifuged at 1500× *g* for 6 min to get the pellet. The pellets were sequentially suspended (Dulbecco’s modified Eagle’s medium supplemented with 15% fetal bovine serum, Hyclone, Logan, UT, USA), filtered (70-μm-mesh sieve, BD Falcon, New York, NY, USA), centrifuged (800× *g* for 5 min) to isolate MuSCs. Using a Percoll gradient (90, 40, and 20%) (Sigma-Aldrich, Shanghai, China), we enriched cells between 40% and 90%. Then the enriched MuSCs were subsequently validated by immunostaining with an antibody for PAX7 (rabbit anti-PAX7, 1:100 dilution, Boster, Wuhan, China), a critical marker for MuSCs. Finally, the purified PAX7+ MuSCs were stored in liquid nitrogen.

### 4.4. Cell Culturing and Transfecting

Cells were, in general, cultured in a 5% CO2 atmosphere at 37 °C. MuSCs were initially seeded in 6-well (~2 × 10^5^ cells per well) or 12-well (~1 × 10^5^ cells per well) plates in growth medium (GM) that contained DMEM plus 10% fetal bovine serum (Gibco, Grand Island, NY, USA) and 2% Penicillin and Streptomycin (Invitrogen, Waltham, MA, USA) solution. When cells were at 80–90% confluency, GM was replaced by differentiation medium (DM) via reducing FBS from 10% to 2% to induce differentiation. The medium was refreshed every 2 d.

For the gain and loss function study, when cells were at 80~90% confluence, the GM was replaced with DMEM supplemented with 10% FBS. Then, cells were transfected using Lipofectamine 3000 (Invitrogen, Waltham, MA, USA) with interfering RNA (siCtrl, siIGF2BP1, in-Ctrl, or in-193b-3p) or overexpression plasmid (pCtrl or pIGF2BP1) and chi-miR-193b-3p mimic (mi-193b-3p, RiboBio, Guangzhou, China) at indicative concentrations, according to the manufacturer’s direction. The transfected cells were kept in GM or replaced with DM. Cells were collected for RNA/protein assay at 48 h, stained with EdU at indicative time points, or with MyHC (Myosin Heavy Chain) antibody at 7th d post differentiation.

Given that culturing cells are easily contaminated by mycoplasma, we monitored mycoplasma contamination based on previously reported protocols [57]. Briefly, cell culture supernatant was collected and detected using TransDetect^®^ PCR Mycoplasma Detection Kit (TRansgen Biotech, Beijing, China) to ensure cells were mycoplasma-free before transfection (Appendix A).

### 4.5. Reporter Constructs and Luciferase Assays

To evaluate the base-pairing relationship between IGF2BP1 and miR-193b-3p, we predicted the potential MRE (miRNA recognition element) of miR-193b-3p within 3′ UTR regions of IGF2BP1 mRNA with online software FINDTAR3 Version 1.0 (http://bio.sz.tsinghua.edu.cn/, accessed on 15 October 2019) and BIBIServ2 (https://bibiserv.cebitec.uni-bielefeld.de/rnahybrid/, accessed on 15 October 2019). Then a 470 bp-length 3′ UTR region of IGF2BP1 mRNA containing the wild-type (wt, GGCCGG) or mutant (mut, GTAAAG) were synthesized and subcloned into the psiCHECK-2 vector (Promega, Madison, MI, USA) to generate IGF2BP1-wt and IGF2BP1-mut vectors (Sangon Biotech, Shanghai, China), respectively.

To detect the enhancer activity of miR-193b-3p, two fragments harboring goat miR-193b-3p or 5p, respectively, e.g., pre-193b-3p (5′-ggtaccGTTTATGTTTTATCCAACTGGCCCACAAAGTCCCGCTTTTGGGGTCATTCTAGACGGCGAGGGATTCAGctcgag-3′) and pre-193b-5p (5′-ggtaccGGAGGCTGTGGTCCCAGAATCGGGGTTTTGAGGGCGAGATGAGTTTATGTTTTATCCctcgag-3′) were successfully amplified and subcloned into the pGL3 Promoter vector (Promega, Madison, WI, USA) between the KpnI and XhoI sites. These constructs were cotransfected with pRLT-K (Promega, Madison, WI, USA), encoding the renilla luciferase gene to measure transfection efficiency.

For the luciferase reporter assays, goat MuSCs (~1 × 10^4^ cells per well), mouse C2C12 myoblasts (~1 × 10^5^ cells per well, Thermo Fisher Scientific, Waltham, MA, USA), and HeLa cells (~2 × 10^5^ cells per well, ATCC, Manassas, VA, USA) were cultured in GM on 24-well plates, and cotransfected with vectors (640 ng/well for each vector) using Lipofectamine 2000 reagent when cells reached 80–90% confluence. After transfection for 48 h, the transfected cells were lysed. According to the user’s guidelines, the activities of firefly and renilla luciferase were measured using a dual-luciferase reporter assay system (Promega, Madison, WI, USA).

### 4.6. Overexpressing and Interfering Plasmids Construct

To produce IGF2BP1 and miR-193b-3p overexpressing vector, we amplified the coding sequence (CDS) of IGF2BP1 (XM_005693695.3) (forward primer 5′-cccaagcttaTGAACAAGCTGTACATCGG; reverse primer 5′—ccggaattcTCACTTCCTCCGGGCCTGGGC, 1734 bp) from cDNA, and 371 bp region of the miR-193b-3p gene (forward primer 5′—cccaagcttaCTGTTCTCCCGTCATTCC—3′; the reverse primer 5′—ccggaattcGTAGCAAACCTCCCCTCTT—3′) from the goat genome DNA. Then amplicons were subcloned into the pcDNA 3.1(+) vector separately by using double digestion with HindⅢ and EcoRI, and T4 DNA ligase (Invitrogen, Carlsbad, CA, USA), according to the manufacturer’s guidelines. PCR combining with Sanger sequencing was performed to validate the target gene.

To provide solid interfering results for IGF2BP1, three small interfering RNA, including siRNA1, CAGCTCCTTTATGCAGGCT; siRNA2, CTCCTTTATGCAGGCTCCA; and siRNA3 CTTTATGCAGGCTCCAGAG, targeting goat IGF2BP1 mRNA, were designed and synthesized in RiboBio (Guangzhou, China). We transfected them separately into MuSCs at two concentrations (50 nM and 100 nM) and then quantified IGF2BP1 mRNAs. The results suggest that IGF2BP1 levels are efficiently decreased at 100 nM for each siRNA, thus in the following experiment, we pooled them at 100 nM concentration. In addition, the miR-193b-3p miRNA mimic (mi-193b-3p) and control (mi-Ctrl), as well as inhibitor (in-193b-3p) and control (in-Ctrl) were ordered from RiboBio (Guangzhou, China).

### 4.7. RNA Extraction, Quality, Reverse Transcription, and Real-Time PCR (RT-qPCR)

Following the manufacturer’s instructions, total RNAs were extracted from tissues or cultured cells using RNAiso Plus reagent (Takara, Dalian, China). To reduce DNA contamination, RNA samples were treated with DNase twice. Then, we measured their absorbances at 230, 260, and 280 nm using a Nano One spectrophotometer (Thermo Fisher Scientific, Waltham, MA, USA). Additionally, samples with A260/A230 and A260/A280 ratio between 1.9 and 2.0 were kept, followed by monitoring their integrity on 1.5% agarose gel electrophoresis and concentration on a NanoDrop 2000c Spectrophotometer (Thermo-Fisher Scientific, Waltham, MA, USA).

The qualified RNAs (~1 mg) were reversely transcribed into cDNA using the PrimeScript™ RT reagent Kit with gDNA Eraser or miRNA PrimeScript RT reagent Kit (Takara, Dalian, China) separately for mRNA or miRNA assay. Then, according to the manufacturer’s guide, expression levels of target genes in these cDNA were accurately measured by using real-time PCR (RT-qPCR) on the Bio-Rad CFX96 system (Bio-Rad, Shanghai, China) with SYBR Premix Ex TaqTM II (Takara, Dalian, China). Each treatment was performed at least three independent times, and each sample triplicated in qPCR. Moreover, to enhance the accuracy, three housekeeping genes in goat (ACTB, Actin Beta; SDHA, Succinate Dehydrogenase Complex Flavoprotein Subunit A; PGK1, Phosphoglycerate Kinase 1) and mouse (ACTB; GAPDH; Hprt, Hypoxanthine Phosphoribosyltransferase 1) were used as an internal control. The 2^−ΔΔCt^ or 2^−ΔCt^ methods were employed to calculate the relative RNA levels of target genes.

### 4.8. Primers Designing and Validation

Primer pairs were designed using Primer Premier 5.0 and their specificities were screened via Primer-BLAST (https://www.ncbi.nlm.nih.gov/, accessed on 21 March 2019) with both goat and human Refseq as background, and then synthesized and PAGE purified (Sangon, Shanghai, China). Moreover, we validated their specificity using Sanger sequencing and quantified the amplification efficiency of each RT-qPCR primer pair (between 95~105%). The detailed information for primers was listed in Appendix A.

### 4.9. Western Blotting and Immunofluorescence for Protein Analyses

To analyze protein levels of the target genes via western blotting (WB), we extracted total proteins in lysed cells by using radioimmunoprecipitation assay (RIPA) (Beyotime, Shanghai, China). After quantification through the bicinchoninic acid (BCA) method (Beyotime, Shanghai, China), protein samples (~20 μg) were sequentially loaded for electrophoresis, transferred on polyvinylidene fluoride (PVDF) membranes (Millipore, Burlington, VT, USA), incubated with primary antibodies for IGF2BP1, MyoG, MyoD, and PAX3 (Abcam, Cambridge, UK) at 4 °C overnight and secondary IgG (Beyotime, Shanghai, China) for 2 h. After the addition of horseradish peroxidase (HRP) substrate, protein bands were exposed via electrochemiluminescence (ECL) (Pierce, Appleton, WI, USA) and analyzed by using ImageJ software (vision 2.0). GAPDH (BOSTER, Wuhan, China) worked as a loading control.

For immunofluorescence assay, MuSCs (seeded in 3.5-cm Petri dishes with ~2 × 10^4^ cells per dish) cultured in DM for 7 days were fixed with 4% paraformaldehyde (room temperature, 15 min), washed with 1 mL PBS (3 times), permeabilized with 1 mL 0.5% Triton X-100 (4 °C, 10 min), blocked in 1 mL 2% bovine serum albumin (37 °C, 30 min), incubated with anti-mouse Myosin heavy chain (MyHC) (1: 200, 4 °C, overnight) (R&D Systems, Minneapolis, USA) and secondary antibodies Cy3_IgG (H + L) (1: 200, Solarbio, Shanghai, China) at 37 °C for 2 h sequentially. Finally, 0.05μg/mL DAPI (4′, 6′-diamidino-2-phenylindole; Invitrogen, Carlsbad, CA, USA)) was added to cells and kept in the dark (37 °C for 10 min). Images were captured using an Olympus IX53 inverted microscope (Tokyo, Japan) and then analyzed using ImageJ software (vision 2.0).

### 4.10. CCK-8 Assay

Cells were inoculated in 96-well plates with 2 × 10^3^ cells per well initially (~30% confluency) and cultured in GM for 2 days, then transfected with different vectors according to protocols described above. Every 24 h, the absorbance value for each sample was measured at 450 nm by using a Microplate Reader (Model 680; Bio-Rad, Hercules, CA, USA), after adding 10 μL of Cell-Counting Kit-8 (CCK-8) reagents (Solarbio, Beijing, China) to the cells for 2 h.

### 4.11. EdU Assays for Cell Proliferation

MuSCs (2 × 10^3^ cells per well) were initially cultured and transfected same as in the CCK-8 assay. Twelve hours after transfection, primary myoblasts were cultured in GM consisting of 10 μM 5-ethynyl-2′-deoxyuridine (EdU; RiboBio, Wuhan, China). Every 24 h, the cells were fixed (4% PFA at room temperature for 30 min), permeabilized (0.5% Triton X-100), incubated (1 × Apollo reaction cocktail for 30 min), and then stained (1 × Hoechst 33342 for 30 min). Finally, we quantified the EdU-stained cells (ratio of EdU+ myoblasts to all) using randomly selected fields captured shortly after staining by employing an Olympus IX53 inverted microscope (Olympus, Tokyo, Japan). Assays were performed at least three times.

### 4.12. Dual-Labeled Fluorescence in Situ Hybridization (FISH)

To detect the spatial localization of IGF2BP1 and miR-193b-3p RNA, the Cy5- labeled fluorescence oligonucleotide probes, complementary to goat IGF2BP1 mRNA (5′-GCCGGATTTGACCAAGAACTGGCCGCTGTAGGA-3′, red), and FAM-labeled chi-miR-193b-3p probes (5′-AGCGGGACTTTGTGGGCCAGTT-3′, green) were synthesized (Servicebio, Wuhan, China). MuSCs proliferating for 2 d and differentiating for 7 d were rinsed with PBS (5 min), fixed in 4% DEPC-treated paraformaldehyde (room temperature, 10 min), permeated in PBS supplemented with 0.5% Triton (4 °C, 5 min), and washed 3 times ×5 min with PBS (PH 7.4). Then, the FISH experiments were performed in a dark environment with the following procedures. For each well, 200 μL pre-hybridized solution (37 °C, 30 min) was added, and hybridized with 8 ng/μL probes in 50 μL buffer (2× SSC, 10% formamide, 10% dextran sulfate) at 37 °C overnight. The cells were sequentially washed with a hybridized solution I, II, and III at 42 °C 3 times × 5 min, 1 time, and 1 time, and finally stained with 0.05 μg/ mL DAPI (Invitrogen, Carlsbad, CA, USA) at 37 °C for 10 min, washed with PBS 3 times × 5 min, and sealed with 25% glycerin.

Images were captured less than 5 h after hybridization, using Nikon DS-U3 in Nikon Eclipse Ti-SR, and then analyzed using the CaseViewer software version 2.3.0.

### 4.13. Nuclear-Cytoplasmic Fractionation

To accurately quantify target genes in the cytoplasm and nucleus separately, MuSCs proliferating for 2 d and differentiating for 7 d were fractionated into nuclear/cytoplasmic parts by using Nuclear/cytoplasmic fractionation and a Nuclear RNA Purification Kit (Norgen Biotek, Thorold, ON, Canada) as described before [58]. In brief, the total RNA was separately extracted from both the supernatant (cytosol fraction) and pellet (nuclei), retro-transcribed into cDNA, and it quantified targeted genes using RT-qPCR. The nuclear-enriched U6 and cytoplasm-enriched 18S were used as controls.

### 4.14. ActD Analysis

According to methods described previously [59], we analyzed the effect of miR-193b-3p on IGF2BP1 mRNA stability by using actinomycin D (ActD, Sigma, Shanghai, China) to block new mRNA synthesis. In general, MuSCs were cultured in 12-well (~1 × 105 cells per well) plates and separately transfected with miR-193b-3p mimic, mimic NC, inhibitor, and inhibitor NC for 24 h. Then, cells were treated with actinomycin D (5 μg/mL) and harvested at 0, 60, and 120 min. Each treatment contained at least three independent biological repetitions. The total RNA was extracted from each sample, and finally the remaining levels of IGF2BP1 mRNA were quantified using the canonical RT-qPCR method.

### 4.15. mRNA-seq and Bioinformatic Analyses

Library preparation and poly(A) selection mRNA-seq was performed at Novogene Company (Beijing, China). Briefly, using NEBNext^®^ UltraTM RNA Library Prep Kit Illumina^®^ (Illumina, San Diego, CA, USA), qualified total RNA samples (200 ng) extracted from cells transfected with miR-193b-3p mimic or control (*n* = 4 per group) were used to isolate the polyA fraction (mRNA), followed by fragmentation and generation of double-stranded cDNA. Libraries were sequentially evaluated by Qubit 2.0 Fluorometer and Agilent 2100 bioanalyzer. Then, qualified libraries were sequenced in Illumina HiSeq 2500 platform (Illumina, San Diego, CA, USA) with a 2 × 150 bp pair-end.

The raw data of each sample were filtered to obtain the clean data (>8 Gb per sample), then about 97.09~97.37% clean reads were quickly and accurately mapped onto the Capra_hircus _ARS1 reference genome (ftp://ftp.ncbi.nlm.nih.gov/genomes/all/GCF/001/704/415/GCF_001704415.1_ARS1/GCF_001704415.1_ARS1_genomic.fna.gz, accessed on 11 July 2019) using HISAT2, among which as many as 93.94~94.3% clean reads were uniquely mapped. The read count for each one was calculated using fragments per kilobase of transcript sequence per million base pairs sequenced (FPKM) value to evaluate gene expression. Differentially expressed genes (DEGs) between samples were canonically identified by DESeq2 R package vision 1.16.1 (|log2(FoldChange)| > 0 and *p*-adj < 0.05) [60].

Transcriptome clustering based on principal component analysis (PCA) indicates that miR_193b-3p mimic or control (mi-Ctrl)-treated cells are distinctly separated. Function Enrichment Analyses of DEGs, including Gene Ontology (GO) enrichment analysis and KEGG pathway, were implemented by using the DESeq2, with *p*-adj < 0.05 (adjusted via Benjamini-Hochberg) considered as significantly enriched. Meanwhile, the top 50 terms for miR-193b-3p-upregulated or -downregulated DE genes were enriched using online software GSEA (https://www.gsea-msigdb.org/gsea/msigdb/index.jsp, accessed on 10 September 2019) based on the Molecular Signatures Database v7.4.

### 4.16. Flow Cytometric Assay

To simply discriminate sub-cycling cells, we performed flow cytometric analysis of cell DNA content using Propidium Iodide (PI)/RNase Staining Buffer (550825, BD Pharmingen™, San Diego, CA, USA). Basically, according to the user’s guide, a total of ~2 × 105 goat MuSCs and C2C12c were seeded in 6-well plates and then treated with miR-193b-3p (mi-193b-3p or mi-Ctrl, 50 nM) or IGF2BP1 (pIGF2BP1 or pCtrl, 2.5 µg/mL) for 48 h at 37 °C. Then, the cells were washed twice with cold PBS, treated with 80% ethanol (24 h), then collected through centrifugation (1500 rpm; 4 °C for 10 min). Subsequently, the cells were stained with 500 μL PI/RNase Staining Buffer for 15 min in the dark. Eventually, PI fluorescence was detected on a BD Accuri™ C6 Plus flow cytometer (BD Biosciences, Franklin Lakes, NJ, USA) equipped with a standard filter (585/40 and 670 LP) and channels (FL3-A-PerCP-A). Data were analyzed using ModFit Flowjo 3.1.

### 4.17. Statistical Analysis

Unless stated otherwise, data presented here are shown as mean ± MSE from at least triplicated independent samples or animals. The unpaired two-tailed *t*-test in Graphpad Prism 7.0 was employed to evaluate the means’ difference, with a significant threshold value set at *** *p* < 0.001, ** *p* < 0.01, * *p* < 0.05, and § *p* < 0.1.

## 5. Conclusions

We reported that miR-193b-3p induces proliferation and differentiation of goat MuSCs, phenocopying IGF2BP1 overexpressing. The enhancer-overlapped miRNA-193b-3p carries a dual function ability: activating its target gene within the nucleus, as well as repressing mRNA in the cytoplasm by directly targeting the 3′ UTR of the IGF2BP1. Our research further extends the repertoire of miRNA functions.

## Figures and Tables

**Figure 1 ijms-23-15760-f001:**
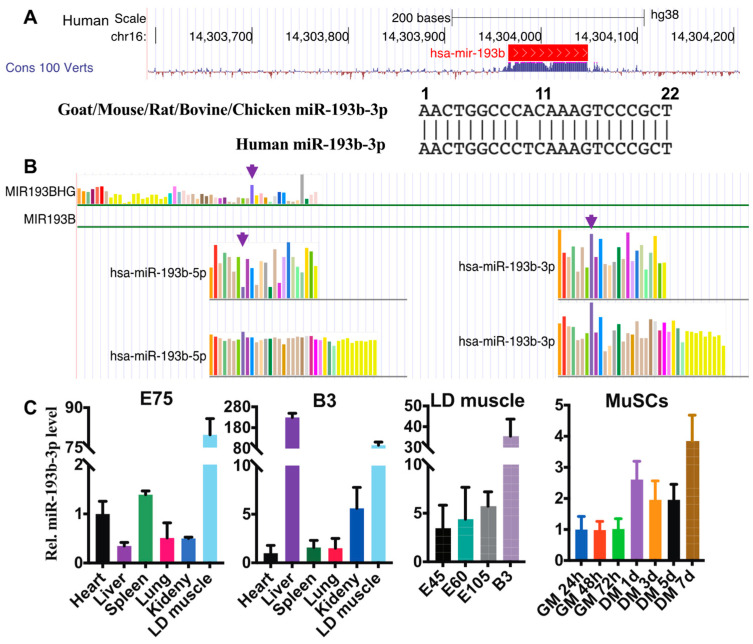
The sequence and expression conservation of miR-193b-3p. (**A**) miR-193b-3p sequence is highly conserved among 100 verts in the UCSC genome browser (upper); the 10th nucleotide of human miR-193b-3p is different from that of other animals (low). (**B**) Expression profiles of human MIR193BHG, miR-193b-5p, and miR-193b-3p in multiple tissues from GTEx. Arrows mark the skeletal muscle. (**C**) Levels of miR-193b-3p in tissues from embryonic 75 days (E75) and 3 d postnatal (B3) goat kids, or longissimus dorsi (LD) muscle from E45 to B3, as well as in cultured skeletal muscle satellite cells (MuSCs). RNA levels are quantified by employing RT-qPCR and calculated by the 2^−ΔΔCt^ methods, with normalization to β-actin (ACTB) and values of the first tissue or timepoint set to 1. Data are shown as mean ± MSE.

**Figure 2 ijms-23-15760-f002:**
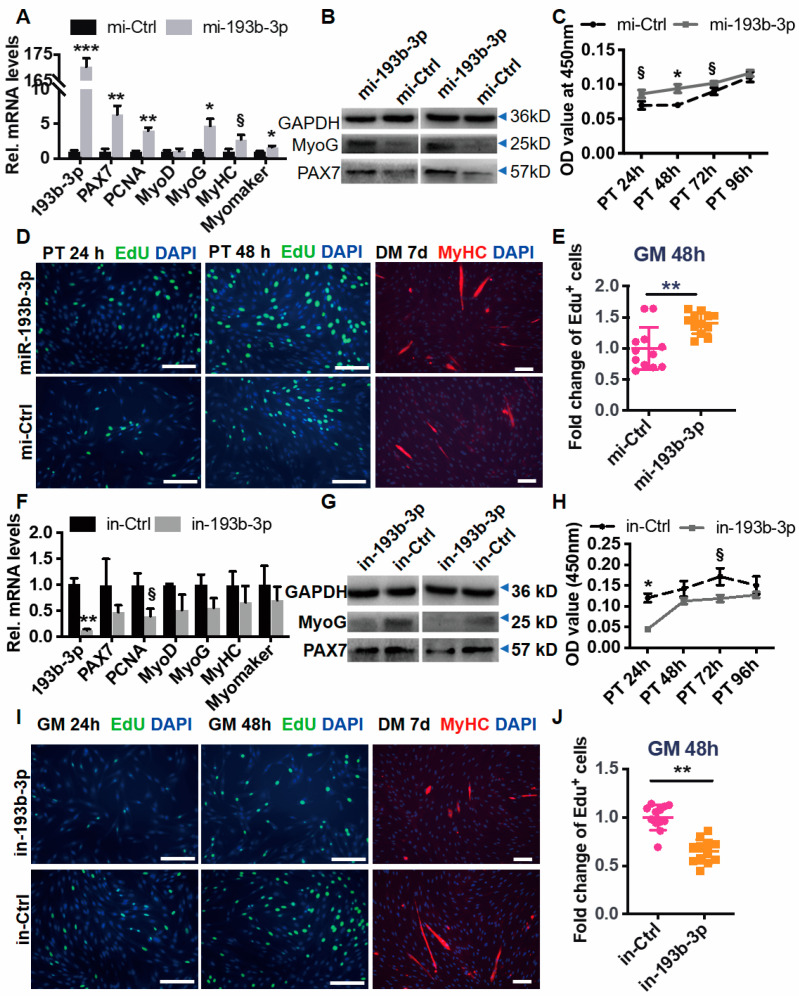
miR-193b-3p promotes the proliferation and differentiation of goat skeletal muscle satellite cells (MuSCs). (**A**) Ectopic miR-193b-3p elevates the mRNA abundance of myogenic genes. The cultured goat MuSCs were transfected with miR-193b-3p mimic or control (mi-193b-3p or mi-Ctrl, 50 nM). The RNA levels are quantified via RT-qPCR and calculated using the 2^−ΔΔCt^, with β-actin (ACTB) as an internal control and values of the mi-Ctrl set to 1. (**B**) miR-193b-3p upregulates the proteins of PAX7 and MyoG. Western blotting (WB) assay was typically performed to detect MyoG and PAX7 protein in MuSCs treated with miR-193b-3p or mi-Ctrl. GAPDH works as a loading control. (**C**) miR-193b-3p promotes cell proliferation measured using Cell-Counting Kit-8 (CCK-8). (**D**,**E**) miR-193b-3p increases EdU^+^ cells and myotube formation. Left panel, Representative immunofluorescence images of EdU and MyHC staining cells after transfected mi-193b-3p or mi-Ctrl (50 nM). Cells are cultured in GM consisting of 10 μM EdU, stained with anti-DAPI (blue) and anti-MyHC (red). Scale bar = 100 μm. Right panel, EdU-stained cells (ratio of EdU^+^ myoblasts to all) are evaluated using randomly selected fields and normalized to control. Each treatment is at least tripled. (**F**) Deficiency of miR-193b-3p suppresses the mRNA abundance of myogenic genes. The RNA levels are quantified via RT-qPCR and calculated using the 2^−ΔΔCt^, with β-actin as an internal control and values of the mi-Ctrl set to 1. (**G**) Proteins of MyoG and PAX7 downregulated by deficiency of miR-193b-3p. Western blotting (WB) assay is typically performed to detect protein levels of MyoG and PAX7 in MuSCs treated within-193b-3p or in-Ctrl. GAPDH works as a loading control. (**H**) Deficiency of miR-193b-3p delays cell proliferation. (**I**) Representative images of EdU and MyHC staining cells transfected with in-193b-3p or in-Ctrl (50 nM). Cells are cultured in GM consisting of 10 μM EdU, stained with anti-DAPI (blue) and anti- MyHC (red). Scale bar = 100 μm. (**J**) The EdU-stained cells are evaluated using randomly selected fields and normalized to control. Each treatment is at least tripled. An unpaired two-tailed *t*-test is used to evaluate the means difference. Data are shown as mean ± MSE. *** *p* < 0.001, ** *p* < 0.01, * *p* < 0.05, and § *p* < 0.1.

**Figure 3 ijms-23-15760-f003:**
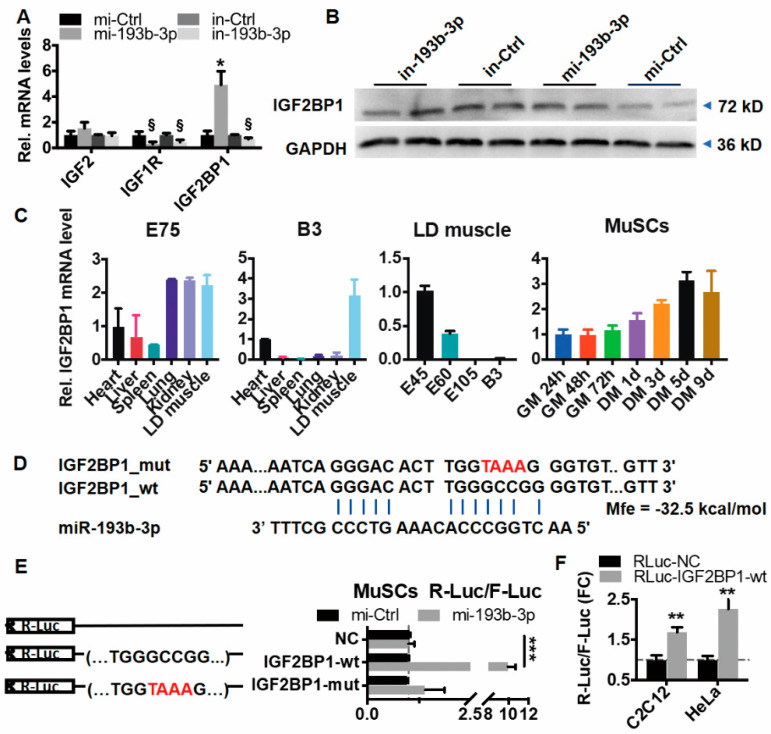
miR-193b-3p induces IGF2BP through a seed-sites match. (**A**) miR-193b-3p promotes IGF2BP1 transcripts. (**B**) miR-193b-3p elevates IGF2BP1 protein. Western blotting (WB) assay was performed to detect IGF2BP1 protein. GAPDH works as a loading control. (**C**) The expression profile of IGF2BP1 in goat tissues and cells. (**D**) miR-193b-3p targets 3′ UTR of IGF2BP1. The potential seed match between miR-193b-3p within 3′ UTR regions of IGF2BP1 is predicted. The blue line marks the complementary nucleotide. Mfe indicates free energy between IGF2BP1-wt and miR-193b-3p. (**E**) The addition of miR-193b-3p activates luciferase activity of wild-type (R-Luc-IGF2BP1-wt) in MuSCs. (**F**) miR-193b-3p activates luciferase activity of R-Luc-IGF2BP1-wt in HeLa and mouse C2C12 myoblast. Data are shown as mean ± MSE. An unpaired two-tailed *t*-test is used to evaluate the means difference. *** *p* < 0.001, ** *p* < 0.01, * *p* < 0.05, and § *p* < 0.1.

**Figure 4 ijms-23-15760-f004:**
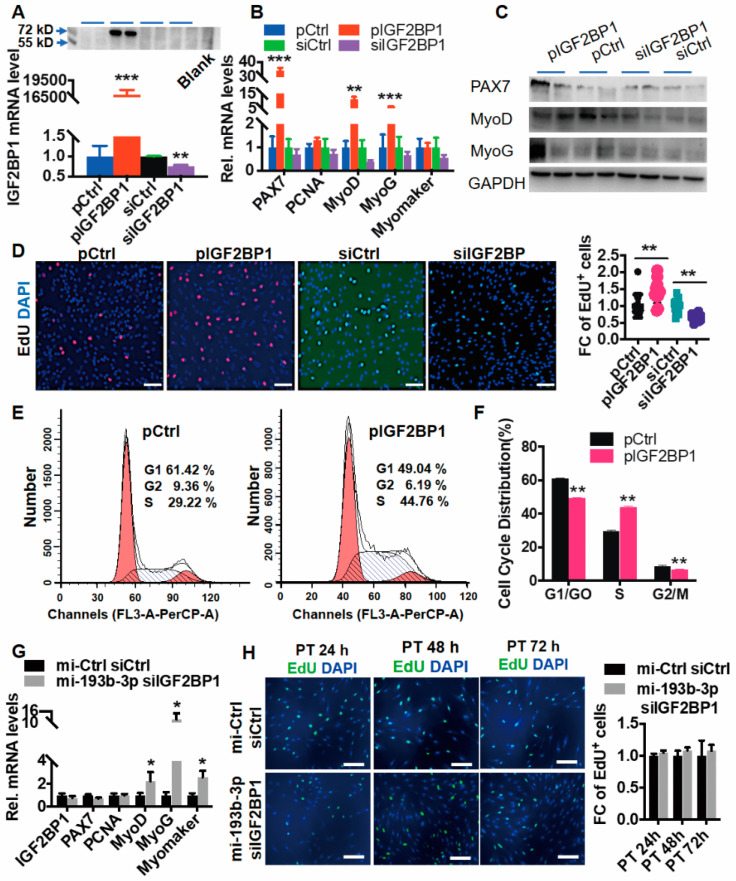
IGF2BP1 promotes myoblast proliferation. (**A**) Levels of IGF2BP1 are successfully disturbed. mRNA (down panel) and protein (upper panel) levels of IGF2BP1 were measured in MuSCs 48 h post-transfection of pIGF2BP1 (3 μg/mL) or siIGF2BP1 (100 nM). (**B**,**C**) Effect of IGF2BP1 on myogenic gene RNA and protein levels. (**D**) IGF2BP1 elevates cell proliferation. Representative immunofluorescence images of EdU staining cells treated with pCtrl and pIGF2BP1 (3 μg/mL, red) or siCtrl and siIGF2BP1 (100 nM, green), cultured in GM consisting of 10 μM EdU (red or green) for 48 h, stained with anti-DAPI (blue). Scale bar = 100 μm. The fold change of EdU^+^ cells is evaluated using randomly selected fields and normalized to control. (**E**,**F**) Cell cycle is affected by IGF2BP1 alteration. Flow cytometric assay is performed for cells treated with pIGF2BP1 or pCtrl (*n* = 3). (**G**) Interfering IGF2BP1 retards miR-193b-3p-induced myogenic proliferation. Total RNA was extracted in cells cotransfected with miR-193b-3p mimic (mi-193b-3p, 50 nM) and interfering RNA against IGF2BP1 (siIGF2BP1, 100 nM) for 48 h in growth media (GM) or differentiated for 48 h (DM). RNA levels of target genes were quantified via RT-qPCR and calculated using the 2^−ΔΔCt^. (**H**) Representative immunofluorescence images of EdU and DAPI (blue) staining cells post-transfection for 24, 48, and 72 h. Scale bar = 100 μm. The fold change of EdU^+^ cells is evaluated using randomly selected fields and normalized to control. Data are shown as mean ± MSE. An unpaired two-tailed *t*-test is used to evaluate the means difference. *** *p* < 0.001, ** *p* < 0.01, and * *p* < 0.05.

**Figure 5 ijms-23-15760-f005:**
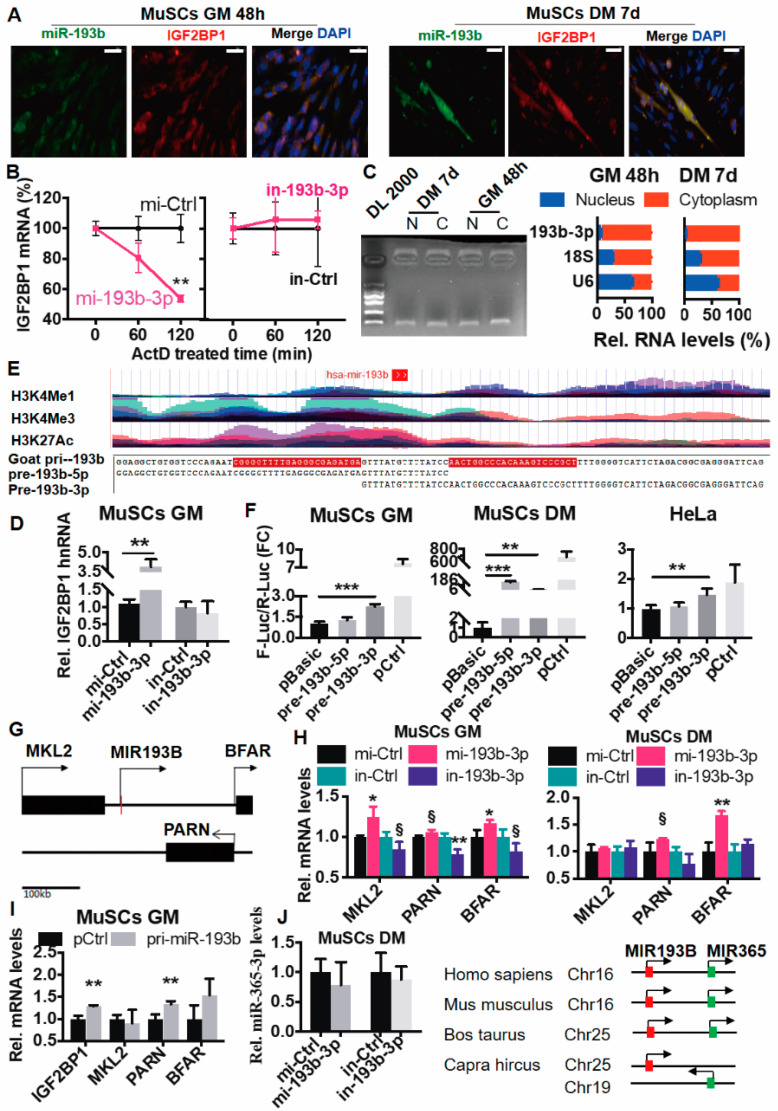
miR-193b-3p activates IGF2BP1 transcriptionally as an enhancer-related miRNA. (**A**) The overlapped signals of miR-193b-3p and IGF2BP1 mRNA. The Cy5-labeled probes target goat IGF2BP1 mRNA (red), and FAM-labeled chi-miR-193b-3p probes (green) are synthesized to localize them in MuSCs. The nucleus is stained with anti-DAPI (blue). Scale bar = 50 μm. (**B**) miR-193b-3p destabilizes IGF2BP1 mRNA. (**C**) miR-193b-3p distributes in both cytoplasm and nucleus. N, Nucleus; C, Cytoplasm. (**D**) miR-193b-3p promotes levels of IGF2BP1 hnRNA. (**E**) Human miR-193b (hsa-mir-193b) is located in the enhancer region. The upper panel, hsa-mir-193b sits in the peak and valley of H3K4Me1/H3K4me3/ H3K27ac derived from 7 cell lines in ENCODE UCSC. Lower panel, the sequence of goat pri-miR-193b and pre-miR-193b-5p and 3p. (**F**) Pre-miR-193b-3p induces gene expression. Promoter vectors including pre-193b-3p, pre-193b-5p as well as the blank (pBasic) and control (pCtrl) (1.28 μg/mL) were transfected into MuSC and HeLa, and F-Luc/R-Luc levels were measured 48 h post transection. (**G**) Neighboring genes of miR-193b-3p, including MKL2, BFAR, and PARN, are the same in the goat and mouse genome. (**H**) Goat miR-193b induces the expression of its neighboring genes. (**I**) Pri-miR-193b activates the abundance of its neighboring genes and IGF2BP1. (**J**) Goat miR-193b-3p is unable to induce miR-365-3p. Left panel, Disturbing miR-193b-3p in goat MuSCs fails to elevate transcripts of miR-365 (right panel). Right panel, miR193b (red box) and miR365a (green box) in humans, mouse, and bovine are neighboring, while in goats they are distributed on different chromosomes. An arrow marks the transcriptional direction. Data are shown as mean ± MSE. An unpaired two-tailed *t*-test is used to evaluate the means difference. *** *p* < 0.001, ** *p* < 0.01, * *p* < 0.05, and § *p* < 0.1.

**Figure 6 ijms-23-15760-f006:**
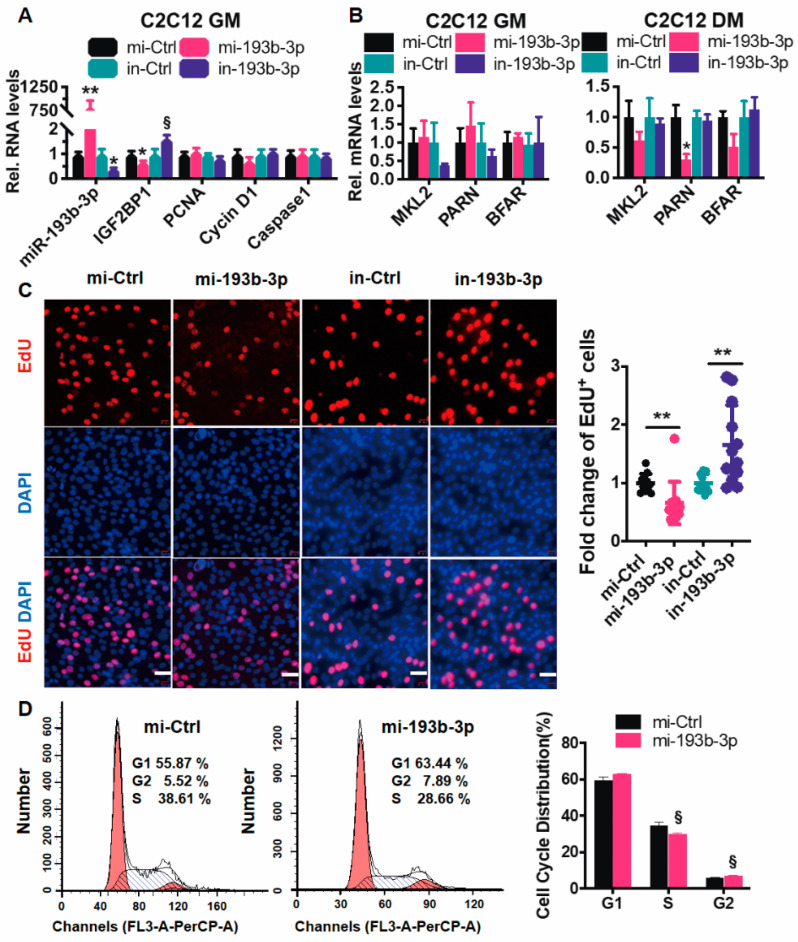
miR-193b-3p negatively regulates mouse IGF2BP1 and suppresses C2C12 proliferation. (**A**,**B**) miR-193b-3p decreases transcripts of IGF2BP1 (**A**) but barely affects the abundance of its neighboring genes in C2C12 (**B**). (**C**) miR-193b-3p deduces C2C12 proliferation. Left panel, Representative immunofluorescence images of EdU staining cells in miR-193b-3p-disturbed C2C12 (mi-Ctrl and mi-193b-3p, in-Ctrl and in-193b-3p, 50 nM). Scale bar = 100 μm. Right panel, Fold change of EdU^+^ cells (ratio of EdU^+^ myoblasts to all) are evaluated using randomly selected fields and normalized to control. Each treatment is repeated ten times. (**D**) Cell cycle is affected by miR-193b-3p mimic. Flow cytometric assay is performed for cells treated with mi-193b-3p or mi-Ctrl (*n* = 3). Data are shown as mean ± MSE. ** *p* < 0.01, * *p* < 0.05, and § *p* < 0.1.

## Data Availability

The accession number for the raw RNA sequencing data reported here is NCBI BioProject: PRJNA665306.

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
