# Peer review of "miR-193b-3p Promotes Proliferation of Goat Skeletal Muscle Satellite Cells through Activating IGF2BP1"

_ijms, 2022, doi:10.3390/ijms232415760_

Round 1

Reviewer 1 Report

In the manuscript entitled “miR-193b-3p Promotes Proliferation of Goat Skeletal Muscle Satellite Cells through Activating IGF2BP1” Li et al. described the involvement of miR-193b-3p in goat myogenesis.

This interesting novel study presents sound data on the species-specific effect of miR-193b-3p on goat myogenesis. The authors used the gain and loss of function approaches to demonstrate the involvement of miR-193b-IGF2BP1 interaction in the proliferation and differentiation of goat myoblast.

It is a well-written manuscript that delivers clear messages.

It will be beneficial if the author comments on why at the same time, overexpression of  miR-193b-3p, induces myogenic genes that are characteristic for proliferation and differentiation. It is generally accepted that when Pax7 goes down, myogenin and MyHC go up, and vice versa. It is unusual that a factor induces proliferation and myogenic differentiation at the same time.

My other comment is on the opposite effect of miR-193b-3p on goat and mouse myoblasts. This might be because the goat myoblasts are primary cells and C2C12 (mouse myoblasts) is an immortalized cell line. The authors should be more careful in the interpretation of their findings. 

Reviewer 2 Report

The study by Li Li et al., focuses on molecular aspects of goat skeletal muscle, in particular myogenesis-related networks.

The authors showed a relationship between miR-193b-3p and the RNA-binding protein IGF2BP1. They illustrate experiments performed using skeletal muscle satellite cells (MuSCs) obtained from the longissimus dorsi (LD) muscles of neonatal goats. In this cellular system, when over-expressed, miR-193b-3p targets the 3ʹUTR of IGF2BP1 causing its up-regulation at both the transcript and protein level. In this way, miR-193b-3p promotes IGF1BP2-mediated myogenesis of goat skeletal muscle.

A critical aspect of this study is the strong discrepancy between the data obtained on the goat muscle and those obtained in the mouse model (C2C12 cells). This impact negatively on the work and, also, the authors do not produce a possible explanation for this.

In my opinion, the manuscript appears not suitable for publication on IJMS.

Overall, substantial revisions are needed.

-          In many cases panels in figures are not clearly illustrated. For example, the fig 2B needs to specify what it illustrates (a duplicate?)

-          In fig 2D, the immunostaining of MyHC in DM 7 visualizes a low number of myotubes, indicating a defective muscle differentiation in each sample.

-          Line 185: the sentence: “miR-193b-3p potentially activates IGF2BP1 and consequently promotes myogenic proliferation” is not fully supported by the data.

-          Line 238: “S-phase” is the correct text

-          Fiure 5A: FISH experiments, performed to visualize a co-localization between miR-193b-3p and IGF2BP1 mRNA, do not provide quality images, and in my opinion both FISH staining could generate a not specific signal.

Round 2

Reviewer 2 Report

The authors substantially improved their manuscript. However, regarding the issue below I am strongly puzzled.

Authors: Thanks a lot for your suggestion. Although these images are not good enough to present a ‘specific signal,’ it is evident that IGF2BP1 mRNA and miR-193b are dominantly distributed in the cytoplasm, consistent with the results shown in Fig 5C. Therefore, we believe these images of RNA FISH roughly visualize the colocalization and successfully work as a piece of evidence to support the interaction between miR-193b-3p and IGF2BP1 mRNA.

The fact of that two specific mRNAs localize in both the cytoplasm and nucleus, does not mean that they interact with each other. Unfortunately, FISH images do not help to demonstrate a colocalization between miR-193b-3p and IGF2BP1 mRNA.

Instead, images in Fig.5A are indicative of overlapped signals. In this case, it would be more appropriate to use a confocal microscope or implement FISH experiments with RNA-immunoprecipitation (RIP) assay.

Round 3

Reviewer 2 Report

Regarding the figure 5 (Fish experiment), the text "overlapped signals" is now correct and well supported by the images. I accept the manuscript in present form.